# Task-specific roles of local interneurons for inter- and intraglomerular signaling in the insect antennal lobe

Debora Fusca, Peter Kloppenburg*

Biocenter, Institute for Zoology, and Cologne Excellence Cluster on Cellular Stress Responses in Aging-Associated Diseases (CECAD), University of Cologne, Cologne, Germany

**Abstract** Local interneurons (LNs) mediate complex interactions within the antennal lobe, the primary olfactory system of insects, and the functional analog of the vertebrate olfactory bulb. In the cockroach *Periplaneta americana*, as in other insects, several types of LNs with distinctive physiological and morphological properties can be defined. Here, we combined whole-cell patch-clamp recordings and $Ca^{2+}$ imaging of individual LNs to analyze the role of spiking and nonspiking LNs in inter- and intraglomerular signaling during olfactory information processing. Spiking GABAergic LNs reacted to odorant stimulation with a uniform rise in $[Ca^{2+}]_i$ in the ramifications of all innervated glomeruli. In contrast, in nonspiking LNs, glomerular $Ca^{2+}$ signals were odorant specific and varied between glomeruli, resulting in distinct, glomerulus-specific tuning curves. The cell type-specific differences in $Ca^{2+}$ dynamics support the idea that spiking LNs play a primary role in interglomerular signaling, while they assign nonspiking LNs an essential role in intraglomerular signaling.

*For correspondence: peter.kloppenburg@uni-koeln.de

Competing interest: The authors declare that no competing interests exist.

## Introduction

Local interneurons (LNs) with markedly different functional phenotypes are crucial for odor information processing in the insect antennal lobe (AL). The AL is the first synaptic relay in the insect olfactory system, showing striking structural and functional similarities to the vertebrates' olfactory bulb. In many regards, the LNs in the AL are the functional equivalent of granule cells, but also periglomerular and short axon cells in the vertebrate olfactory bulb (*Ennis et al., 2015*; *Shepherd et al., 2004*). They help to structure the odor representation in the AL, ultimately shaping the tuning profiles of the olfactory projection (output) neurons.

Based on initial studies, LNs originally have been characterized as GABAergic and multiglomerular (*Distler, 1989*; *Hoskins et al., 1986*; *Waldrop et al., 1987*). Typically, they can generate $Na^+$-driven action potentials (*Chou et al., 2010*; *Christensen et al., 1993*; *Husch et al., 2009a*; *Seki et al., 2010*) or $Ca^{2+}$-driven spikelets (*Laurent and Davidowitz, 1994*). Accordingly, these neurons have been associated with inhibitory interglomerular signaling, that is, with mediating lateral inhibition to enhance contrast and to control timing and synchronization of neuronal activity (*Assisi and Bazhenov, 2012*; *Assisi et al., 2011*; *Christensen et al., 1998*; *Fujiwara et al., 2014*; *Hong and Wilson, 2015*; *MacLeod and Laurent, 1996*; *Nagel and Wilson, 2016*; *Olsen and Wilson, 2008*; *Sachse and Galizia, 2002*; *Wilson, 2013*). Subsequent studies showed that LNs can also synthesize other potential neurotransmitters and neuromodulators (*Berg et al., 2007*; *Chou et al., 2010*; *Das et al., 2011*; *Distler, 1990*; *Fusca et al., 2013*; *Fusca et al., 2015*; *Neupert et al., 2012*; *Shang et al., 2007*). In fact, they can be excitatory, distributing excitatory synaptic input to (projection) neurons in other glomeruli by chemical and electrical synapses (*Assisi et al., 2012*; *Das et al., 2017*; *Huang et al., 2010*; *Olsen et al., 2007*; *Shang et al., 2007*; *Yaksi and Wilson, 2010*).

Furthermore, nonspiking LNs with weak active membrane properties that do not generate Na⁺-driven action potentials have been described in both holo- and hemimetabolous insect species (*Fuscà and Kloppenburg, 2021*; *Husch et al., 2009a*; *Husch et al., 2009b*; *Tabuchi et al., 2015*). While their functional role for odor information processing is not clear yet, it is plausible to assume that they are functionally highly relevant since they have been found across different insect species.

In *Periplaneta americana*, two main LN types, each with distinct functional properties, have been identified: spiking type I LNs and nonspiking type II LNs (*Fusca et al., 2013*; *Fuscà and Kloppenburg, 2021*; *Husch et al., 2009a*; *Husch et al., 2009b*). Spiking type I LNs generate Na⁺-driven action potentials upon odor stimulation and have a multiglomerular branching pattern with ramifications in many, but not all, glomeruli. The density and pattern of ramifications vary between glomeruli from very dense to sparse, indicating a polarity with defined synaptic input and output regions. All type I LNs are GABAergic (*Distler, 1989*; *Husch et al., 2009a*). The nonspiking type II LNs do not have voltage-activated transient Na⁺ currents and cannot generate Na⁺-driven action potentials. They have comparatively large voltage-dependent Ca²⁺ conductances, which play a crucial role in mediating their active membrane properties (*Fusca et al., 2013*; *Husch et al., 2009a*; *Husch et al., 2009b*). Type II LNs consist of two main subtypes, type IIa and type IIb LNs. Type IIa LNs typically respond with complex, odor-specific patterns of excitation that can include periods of inhibition. Type IIa LNs can be further subdivided into type IIa1 and type IIa2. Type IIa1 LNs are cholinergic and can generate Ca²⁺-driven spikelets on top of odor-induced depolarizations (*Fusca et al., 2013*; *Neupert et al., 2018*). Type IIa2 LNs do not generate spikelets, and their primary transmitter is not known. Type IIb LNs have weak active membrane properties and respond to odor stimulation with sustained, relatively smooth depolarization. Their primary transmitter is unknown. All type II LNs are omniglomerular, with branches in virtually all glomeruli. The density and pattern of arborizations are similar in all glomeruli of a given type II LN but vary between different type II LNs (*Husch et al., 2009a*). In type IIa LNs, the ramifications are similar and evenly distributed over each glomerulus of an individual neuron. On the other hand, in type IIb neurons, the branches cover only parts of each glomerulus, often in a specific layer (*Husch et al., 2009b*).

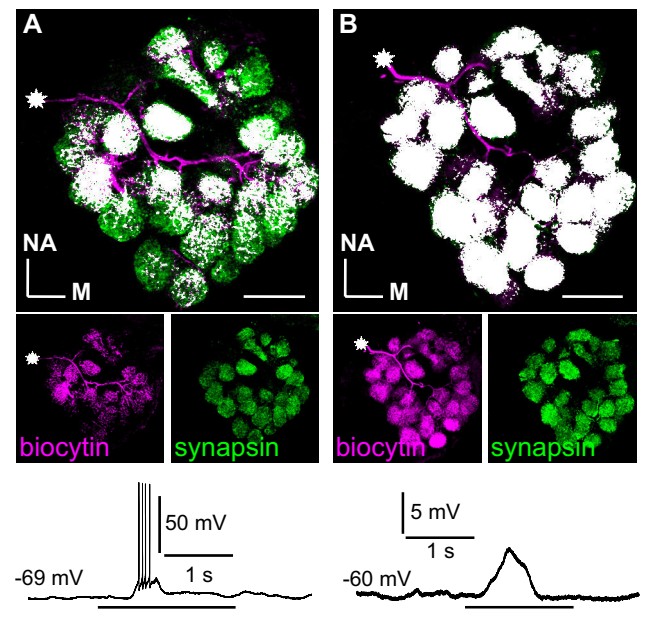

**Figure 1.** Branching patterns and odorant responses of spiking and nonspiking local interneurons. A spiking type I (**A**) and a nonspiking type II local interneuron (**B**) that were labeled with biocytin/streptavidin via the patch pipette. The glomeruli were visualized by synapsin-LIR. (**A**) Type I local interneuron. 13 µm stack of optical sections. The neuron innervates many but not all glomeruli and generates action potentials to an odorant stimulus (benzaldehyde). (**B**) Type II local interneuron. 15 µm stack of optical sections. The neuron innervated all glomeruli and responded to the odorant (benzaldehyde) with a graded depolarization. The stars mark the locations of the somata. Biocytin/streptavidin, magenta; synapsin-LIR, green; double-labeled pixels, white. NA: anterior, M: medial. Scale bars = 100 µm.

Based on their functional and morphological properties, it can be hypothesized that nonspiking type II LNs are primarily involved in intraglomerular signaling since the graded changes in membrane potentials can only spread within the same or electrotonically close glomeruli, as was proposed for nonspiking LNs in the rabbit olfactory bulb (*Bufler et al., 1992b*).

This study's rationale was based on the previously reported structural and functional differences between distinct LN types in the cockroach AL (*Fusca et al., 2013*; *Fusca et al., 2015*; *Fuscà and Kloppenburg, 2021*; *Husch et al., 2009a*; *Husch et al., 2009b*; *Pippow et al., 2009*). Spiking type I LNs are GABAergic, inhibitory, and innervate many but not all glomeruli. While some glomeruli are densely innervated, others are more sparsely or not at all innervated (*Figure 1A*). It has been considered that this reflects an organization with distinctive input and output glomeruli (*Galizia and Kimmerle, 2004*; *Husch et al., 2009a*; *Wilson and Laurent, 2005*). In this model, synaptic input is integrated and triggers action potential firing. The action potentials propagate to the innervated glomeruli and provide a defined glomeruli array with inhibitory synaptic input. Glomeruli can interact independently of their spatial and electrotonic distance. In this scenario, one would expect that odor-evoked glomerular $Ca^{2+}$ signals are dominated by $Ca^{2+}$ influx through voltage-gated channels that are activated by the action potentials. Thus, odor-induced $Ca^{2+}$ signals should be detectable and comparable in all innervated glomeruli.

In contrast, nonspiking type II LNs have very similar branching patterns in all glomeruli (*Figure 1B*, *Husch et al., 2009b*), suggesting that both input and output can occur in every glomerulus. Due to the receptor and sensillum type-specific input configuration of the glomeruli in the AL (*Fujimura et al., 1991*; *Watanabe et al., 2012*), synaptic input during olfactory stimulation typically occurs only in a limited number of glomeruli (*Sachse et al., 1999*; *Silbering et al., 2011*). The resulting stimulus-evoked graded postsynaptic potentials can only spread within the same glomerulus or electrotonically nearby glomeruli. Since these neurons cannot generate $Na^+$-driven action potentials, we hypothesize that the $Ca^{2+}$ signals are dominated by odorant-evoked $Ca^{2+}$ influx through excitatory ligand-gated channels (*Oliveira et al., 2010*).

This study investigated the role of spiking and nonspiking LNs for inter- and intraglomerular signaling during olfactory information processing. To this end, we combined whole-cell patch-clamp recordings with $Ca^{2+}$ imaging to analyze the local $Ca^{2+}$ dynamics of neurites in individual glomeruli as an indicator of signal processing in single LNs. The recordings were performed in the AL of the cockroach *P. americana*. This is an experimental system in which the olfactory system's circuitry has been analyzed in great detail on the physiological (*Bradler C et al., 2016*; *Ernst and Boeckh, 1983*; *Husch et al., 2009a*; *Husch et al., 2009b*; *Lemon, 1997*, *Lemon and Getz, 1998*, *Lemon and Getz, 2000*; *Nishino et al., 2012*, *Nishino et al., 2018*; *Paeger et al., 2017*; *Paoli et al., 2020*; *Pippow et al., 2009*; *Strausfeld and Li, 1999*; *Warren and Kloppenburg, 2014*; *Watanabe et al., 2017*), biochemical (*Distler, 1989*; *Distler, 1990*; *Fusca et al., 2013*; *Fusca et al., 2015*; *Neupert et al., 2012*; *Neupert et al., 2018*), and structural/ ultrastructural levels (*Distler and Boeckh, 1997a*; *Distler and Boeckh, 1997b*; *Distler et al., 1998*; *Malun, 1991a*; *Malun, 1991b*; *Malun et al., 1993*; *Nishino et al., 2015*; *Watanabe et al., 2010*), thus contributing very successfully to understanding olfactory information processing principles.

## Results

LNs of the insect AL are a heterogeneous group of neurons, consisting of different neuronal subpopulations with clearly defined, sometimes fundamentally different functional phenotypes. To study the role of spiking type I LNs and nonspiking type II LNs for inter- and intraglomerular signaling, we performed simultaneous whole-cell patch-clamp recordings and $Ca^{2+}$ imaging in individual neurons. The investigated neuron was labeled with biocytin-streptavidin. This way, the cell types were unequivocally identified by their physiological and morphological characteristics. In the investigated LNs, we measured the intracellular $Ca^{2+}$ dynamics of neurites during olfactory stimulation simultaneously in many individual glomeruli. To determine differences in the odor-induced $Ca^{2+}$ signals between individual glomeruli, tuning curves were constructed from the odor-evoked glomerular $Ca^{2+}$ signals by normalizing them to the maximum signal amplitude of each glomerulus. Overall, this study is based on 23 recordings of type I LNs and 18 recordings of type II LNs. For each recorded LN, between 9 and 25 distinct innervated glomeruli could be identified and individually imaged and analyzed. In the

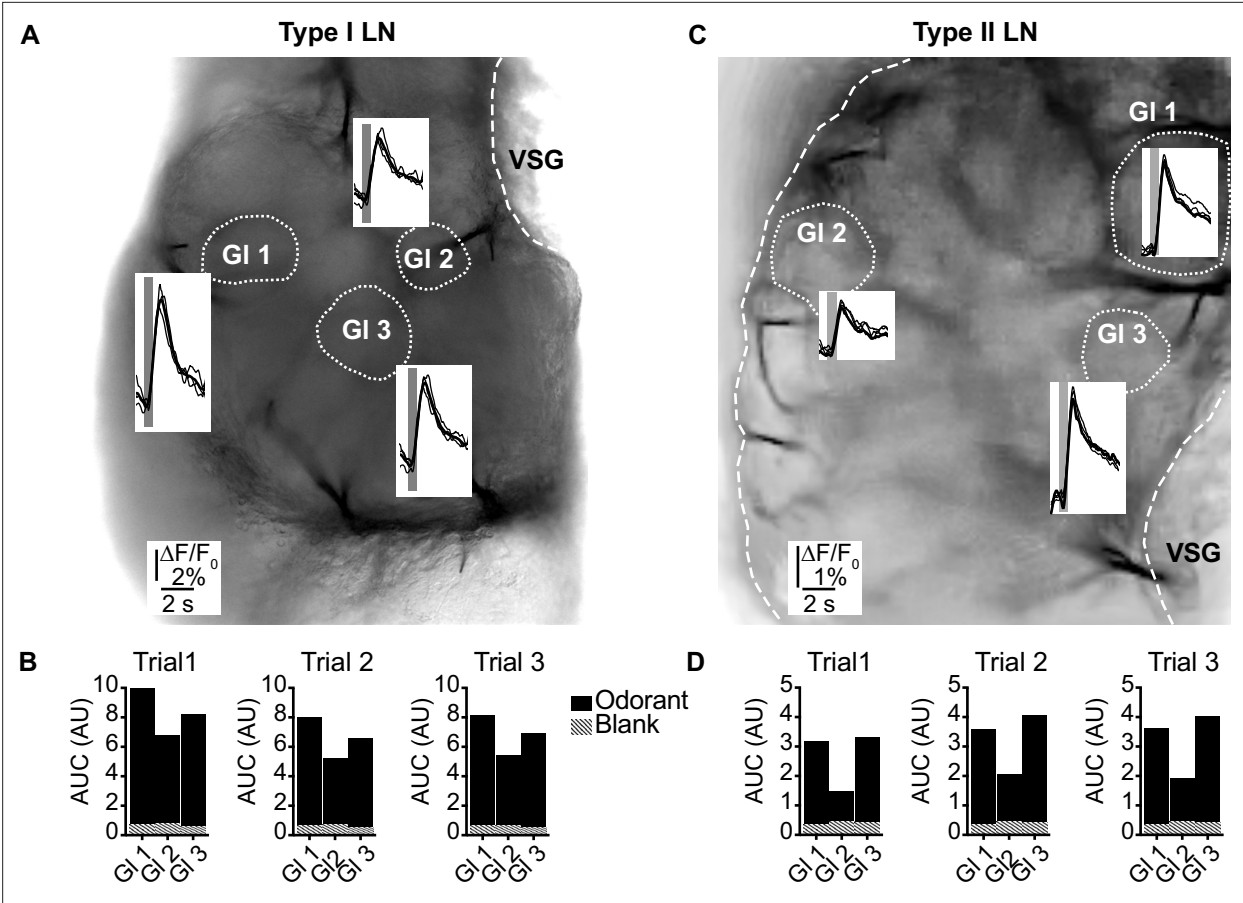

**Figure 2.** (LNI #24; LNII #1) Odorant-induced glomerular calcium signals are reproducible in type I (**A, B**) and type II local interneurons (LNs) (**C, D**). (**A, C**) Transmitted light images of an investigated antennal lobes. The dotted lines mark the recorded glomeruli, and the insets show overlays of Ca²⁺ responses from three trials with the same odorant (hexanol). The gray bars mark the 500 ms odorant stimuli. (**B, D**) Areas under the curves of the Ca²⁺ signals that are shown in (**A**) and (**C**). The first 3 s after stimulus onset were analyzed. Hatched bars represent control signals to blank stimuli. AUC: area under the curve; Gl: glomerulus; VSG: ventrolateral somata group.

The online version of this article includes the following figure supplement(s) for figure 2:

**Source data 1.** Numerical data for **Figure 2B and D**.

legends of **Figure 2**–5, we give the identifier of the analyzed neurons, which can be used to locate the respective data in Individual Neurons-**Source data 1**.

In the first set of experiments, we showed that the odor-induced Ca²⁺ dynamics were highly reproducible when the antennae were repeatedly stimulated with the same odorant (**Figure 2**). This is in line with previous electrophysiological studies, in which LNs responded very reproducibly to repeated olfactory stimulations (**Husch et al., 2009a**; **Husch et al., 2009b**; **Olsen and Wilson, 2008**). Hence, in subsequent experiments, we analyzed single-sweep optophysiological recordings rather than averaged data.

## Uniform glomerular odor responses in spiking type I LNs

All recorded type I LNs displayed characteristic morphological features, that is, arborizations in multiple glomeruli with varying neurite densities between glomeruli (**Figure 3**). Electrophysiologically, type I LNs reacted to odor stimulation of the antennae with odorant-specific patterns of overshooting action potentials (**Figure 3B**). The glomerular Ca²⁺ signals were time-locked with the electrophysiological responses and matched the spike pattern of the electrophysiological recordings. While the absolute amplitudes of the Ca²⁺ signals during a given odorant varied between individual glomeruli, the time course and overall structure of the Ca²⁺ signals were very similar in all recorded glomeruli for a particular odorant (**Figure 3B**, **Figure 3—figure supplement 1**), resulting in identical tuning curves

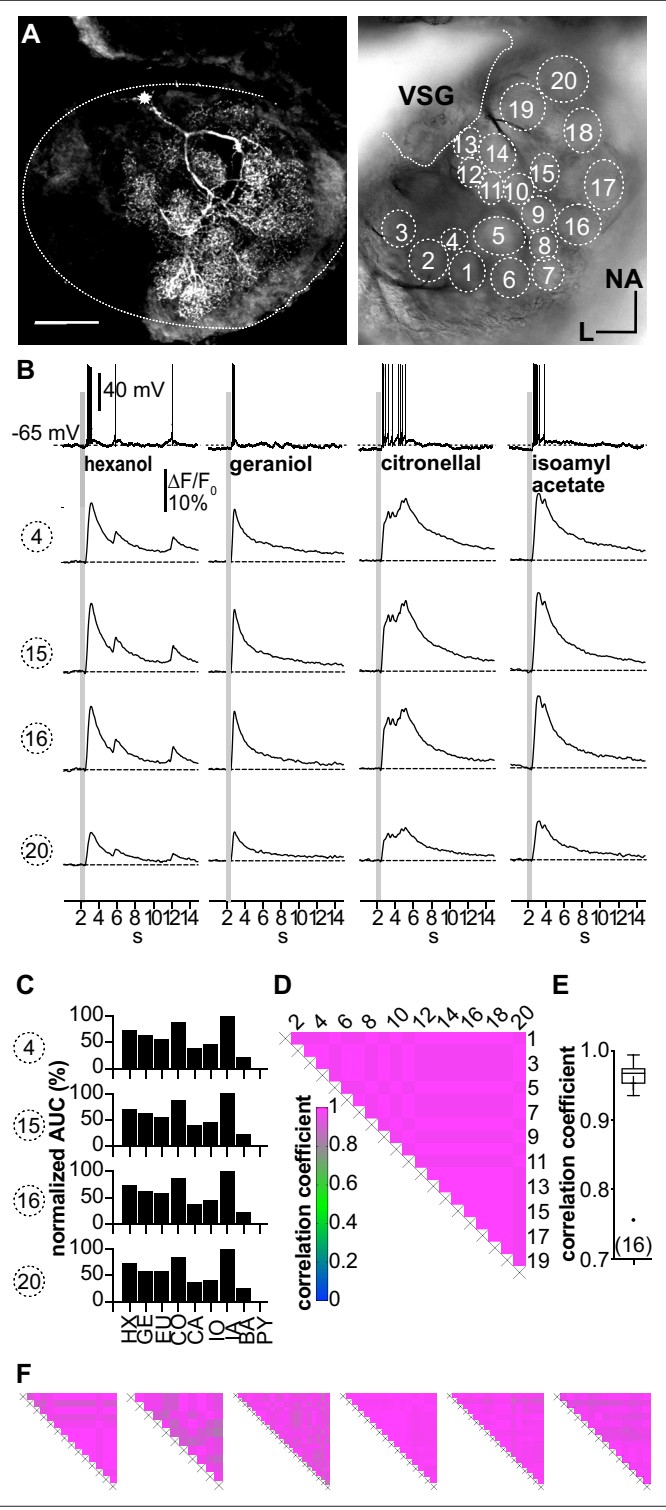

**Figure 3.** (**A–D;** LNI #1) Ca²⁺ imaging in type I local interneurons (LNs) shows uniform glomerular odor responses. (**A**) Left: biocytin/streptavidin-labeled type I LN. The antennal lobe (AL) is outlined by the dotted line. The star marks the position of the soma. Scale bar: 100 µm. Right: transmitted light image of the same AL while the neuron was recorded. Orientation applies to both images. The outlined glomeruli mark the regions of interest (individual glomeruli) that were individually analyzed. (**B**) Electrophysiological responses to four odorants (top traces) and the corresponding Ca²⁺ dynamics of four glomeruli that are marked in (**A**). Gray bars represent the 500 ms odorant stimuli. The neuron responded to different odorants with odorant-specific spike trains. The time courses of the

*Figure 3 continued on next page*

*Figure 3 continued*

Ca²⁺ signals were similar in all glomeruli for a given odorant. (**C**) Tuning curves of glomerular responses. Areas under the curves of the odorant-evoked glomerular Ca²⁺ signals (first 3 s after stimulus onset) were calculated for a set of nine odorants and normalized to the maximum response in the respective glomerulus. Every glomerulus responded most strongly to isoamyl acetate and least to benzaldehyde. (**D**) Heatmap showing the correlations between the glomerular tuning curves of every imaged glomerulus. Numbers correspond to the glomeruli in (**A**). All tuning curves were well correlated with coefficients of ~1 (nonparametric Spearman correlation). (**E**) Mean correlation coefficient across all investigated type I LNs was 0.95 ± 0.06 (N = 16) N values are given in brackets. (**F**) Heatmaps of correlations between glomerular tuning curves from six additional type I LNs (LNI # 7, 8, 12, 14, 15, 18). HX: hexanol; GE: geraniol; EU: eugenol; CO: citronell; CA: citral; IO: ionone; IA: isoamylacetate; BA: benzaldehyde; PY: pyrrolidine.

The online version of this article includes the following figure supplement(s) for figure 3:

**Source data 1.** Numerical data for *Figure 3E*.

**Figure supplement 1.** Ca²⁺ signals from all 20 imaged glomeruli.

for all glomeruli of a given neuron (*Figure 3C*). Accordingly, tuning curves of all imaged glomeruli of a given neuron always correlated with coefficients of ~1, with a mean correlation coefficient across all investigated spiking LNs of $r = 0.95 \pm 0.06$ (N = 16, *Figure 3D–F*).

The large amplitude of the glomerular Ca²⁺ signals, their close match with the electrophysiological recordings, and their correlated dynamics suggest that the Ca²⁺ signals mainly reflect voltage-dependent Ca²⁺ influx induced by propagated action potentials. It seems likely that olfactory inputs from one or a few glomeruli are integrated and trigger action potentials that propagate to the innervated glomeruli, where they induce highly correlated voltage-activated Ca²⁺ signals.

## Suppression of action potential firing in type I LNs decreases correlations of glomerular Ca²⁺ signals

To test whether the observed Ca²⁺ signals in type I LNs mainly reflect action potential-induced Ca²⁺ influx, we used two approaches to prevent the neurons from spiking. The neurons were hyperpolarized to membrane potentials between –80 mV and –100 mV (*Figure 4A*, top trace) or firing was suppressed by intracellularly blocking Na⁺ channels with QX-314. When the generation of action potentials is inhibited, the remaining Ca²⁺ signals should mainly reflect Ca²⁺ influx via ligand-gated channels (e.g., cholinergic receptors, *Oliveira et al., 2010*).

When action potential firing was prevented by hyperpolarization, odor stimulation still elicited Ca²⁺ signals in the glomeruli (*Figure 4A*). Besides a reduction in amplitude, the uniformity of the Ca²⁺ signals between different glomeruli disappeared. In turn, the tuning curves of the individual glomeruli became different from each other (*Figure 4B*, *Figure 4—figure supplement 1*). Quantitatively, this is reflected in the glomerulus-specific odorant responses and the diverse correlations between the glomerular tuning curves, resulting in a decreased mean correlation coefficient across all hyperpolarized type I LNs of $r = 0.75 \pm 0.19$ (N = 6, *Figure 4C–E*). Similar results were obtained when action potential firing was suppressed by intracellularly blocking Na⁺ channels with QX-314 ($r = 0.74 \pm 0.26$, N = 5, *Figure 4D and E*). Differences in mean correlation coefficients were significant between control and hyperpolarized type I LNs (p=0.0073) as well as between control and type I LNs that were treated with QX-314 (p=0.0142). Mean correlation coefficients of hyperpolarized and QX-314-treated type I LNs were not significantly different (p>0.999).

These experiments support the idea that the observed large Ca²⁺ signals are mediated mainly by voltage-gated Ca²⁺ channels, which are activated by the strong depolarizations of the action potentials propagating through the neuron into the innervated glomeruli. These large signals likely mask small local Ca²⁺ signals elicited by ligand-gated (postsynaptic) Ca²⁺ influx into the recorded neuron. Preventing the generation of action potentials selectively in the recorded neurons unmasked small local Ca²⁺ signals, which we interpret as solid evidence for localized excitatory input. Since we observed these signals in more than one glomerulus of a given neuron, we consider this to be physiological evidence that individual type I LNs receive excitatory input not only in one but in multiple glomeruli.

Taken together, our results are in line with the conception that type I LNs integrate and transform their synaptic input to action potential firing to provide inhibitory synaptic input (lateral inhibition) to

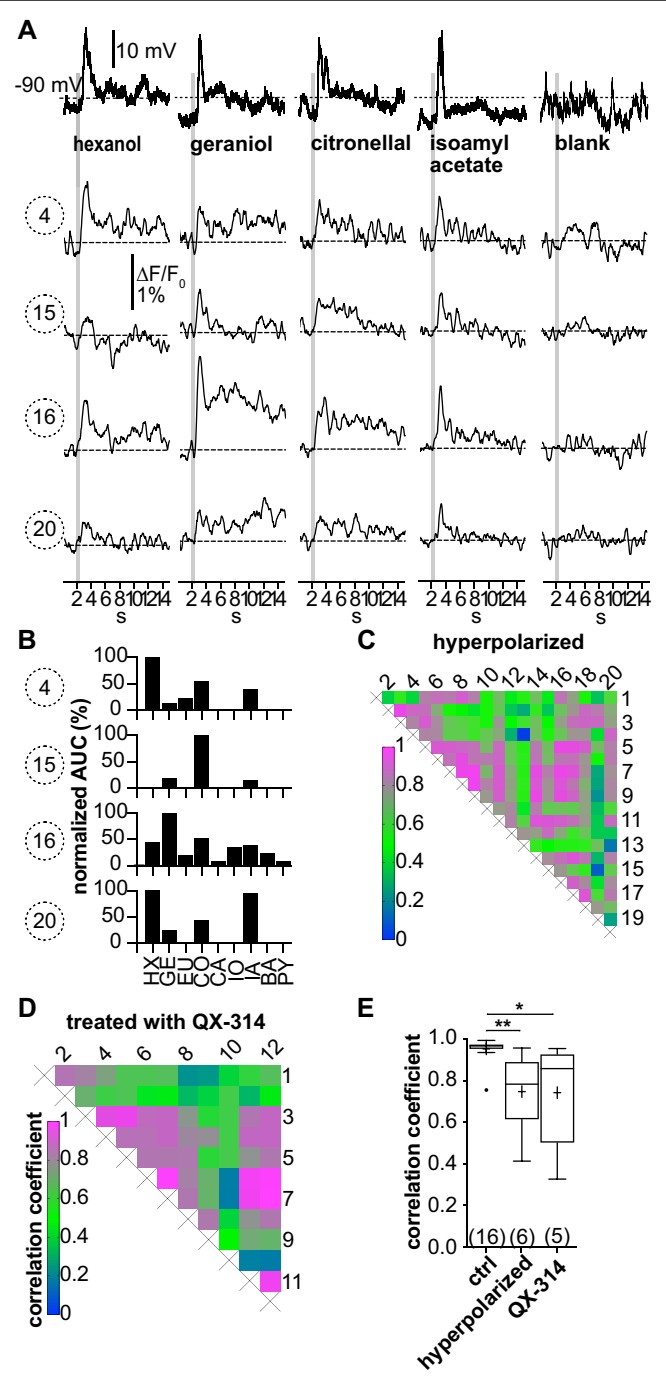

**Figure 4.** (**A–C;** LNI #1) Hyperpolarization below the action potential threshold and pharmacological block of action potential firing prevent the correlation between odorant-induced glomerular Ca²⁺ signals. Data in (**A–C**) are taken from the same type I local interneuron (LN) as in **Figure 3**. (**A**) Electrophysiological responses to odorants (top traces) and corresponding Ca²⁺ dynamics in the same four glomeruli as shown in **Figure 3**. The neuron was hyperpolarized to prevent the generation of action potentials upon stimulation with odorants. Electrophysiologically, the neuron responded with odorant-specific graded depolarizations. The high correlation of the glomerular Ca²⁺ signals shown in **Figure 3** was inhibited. (**B**) The tuning curves of the glomerular responses (for details, see **Figure 3C**) varied considerably, whereby the odorant that triggered the maximum Ca²⁺ signal in each individual glomerulus was different for each glomerulus. (**C**) Heatmap demonstrating the heterogeneous correlations between glomerular tuning curves. Numbers correspond to glomeruli in **Figure 3A**. Correlation coefficients ranged between 0 and 0.95 (median = 0.56). (**D;** LNI #19) Heatmap demonstrating the variable

*Figure 4 continued on next page*

*Figure 4 continued*

correlations between glomerular tuning curves of a neuron that was treated with the intracellular $Na_V$ channel blocker QX-314. Correlation coefficients ranged between 0.2 and 1 (median = 0.7). (**E**) Mean correlation coefficients of hyperpolarized (0.75 ± 0.19, N = 6, p=0.0073) and QX-314-treated (0.74 ± 0.26, N = 5, p=0.0142) type I LNs were significantly decreased compared to the control group (Kruskal–Wallis and Dunn's multiple comparisons test). Hyperpolarized and QX-314-treated type I LNs were not significantly different (p>0.9999). *p<0.05, **p<0.01. N values are given in brackets. Abbreviations as in *Figure 3F*.

The online version of this article includes the following figure supplement(s) for figure 4:

**Source data 1.** Numerical data for *Figure 4E*.

**Figure supplement 1.** Tuning curves of all imaged glomeruli from the neuron shown in *Figure 4A–C*.

---

neurons in a defined array of glomeruli (*Assisi et al., 2011*; *Fujiwara et al., 2014*; *Hong and Wilson, 2015*; *Nagel and Wilson, 2016*; *Olsen and Wilson, 2008*; *Sachse and Galizia, 2002*; *Wilson, 2013*). Our results also provide physiological evidence that an individual type I LN receives excitatory input not only in one but in several glomeruli, which is in line with previous structural- and ultrastructural studies that reported evidence for both pre- and postsynaptic profiles in individual glomeruli (*Berck et al., 2016*; *Distler and Boeckh, 1997b*; *Mohamed et al., 2019*).

## Heterogeneous glomerular odor responses in nonspiking type II LN

In contrast to the uniform $Ca^{2+}$ dynamics during odor stimulation in type I LNs, we observed highly heterogeneous $Ca^{2+}$ dynamics between the individual glomeruli in most type II LNs (*Figure 5*). All recorded type II LNs had the cell type-specific morphology characterized by innervation of all glomeruli with similar neurite densities in all glomeruli of a given neuron (*Figure 5A and E*). All type II LNs typically responded with graded changes in membrane potential to olfactory stimulation, with type IIa1 LNs also able to generate $Ca^{2+}$-driven spikelets (*Figure 5B and F*, top traces). The amplitudes of the corresponding $Ca^{2+}$ signals were in the range of the signals of type I LNs after the suppression of their action potentials. While the electrophysiological responses to different odorants were similar in a given neuron, the corresponding glomerular $Ca^{2+}$ signals were odor specific and varied between glomeruli, resulting in distinct, glomerulus-specific tuning curves (*Figure 5C,G* and *Figure 5—figure supplement 1*), which was also directly evident in a relatively low degree of correlation (*Figure 5I*; *r* = 0.56 ± 0.21, N = 18). However, the correlation between tuning curves of individual glomeruli in a given neuron differed among type II LNs. While in 11 out of 18 nonspiking LNs the majority of glomeruli was individually tuned, in the other 7 neurons, groups of similarly tuned glomeruli were found. This is shown in the heatmaps showing highly correlated $Ca^{2+}$ signals in groups of glomeruli as well as glomeruli that were not correlated (*Figure 5G, H and J*, *Figure 5—figure supplement 1B*). Mechanistically, this could be caused by similar input to several glomeruli (*Watanabe et al., 2012*) or by coordinated activity, for example, via spikelets that were observed in a subtype of nonspiking neurons (type IIa1 LNs, *Fusca et al., 2013*).

## Discussion

Processing of sensory input by networks of spiking and nonspiking interneurons is a common principle in both invertebrate and vertebrate sensory systems, for example, structuring the signal pathway from sensory neurons (tactile hairs) to intersegmental and motor neurons in the insect thoracic ganglion (*Burrows, 1989*; *Pearson and Fourtner, 1975*) and the mammalian olfactory bulb (*Bufler et al., 1992a*; *Bufler et al., 1992b*; *Wellis and Scott, 1990*) and retina (*Diamond, 2017*). Nevertheless, in many systems, the role of nonspiking neurons is not well understood.

LNs are key components of the insect olfactory system. They have fundamentally different functional phenotypes suggesting different tasks during odor information processing. To help elucidate mechanisms of odor processing on the level of individual LNs, this study assessed local $Ca^{2+}$ dynamics in distinct functional compartments (ramifications in individual glomeruli) of spiking type I and nonspiking type II LNs during olfactory information processing. To this end, individual LNs were analyzed by combined whole-cell patch-clamp recordings and $Ca^{2+}$ imaging. Local $Ca^{2+}$ dynamics are likely to reflect the role of LNs in odor information processing, that is, for their potential role in intra- and interglomerular signaling, which depends crucially on signal propagation throughout

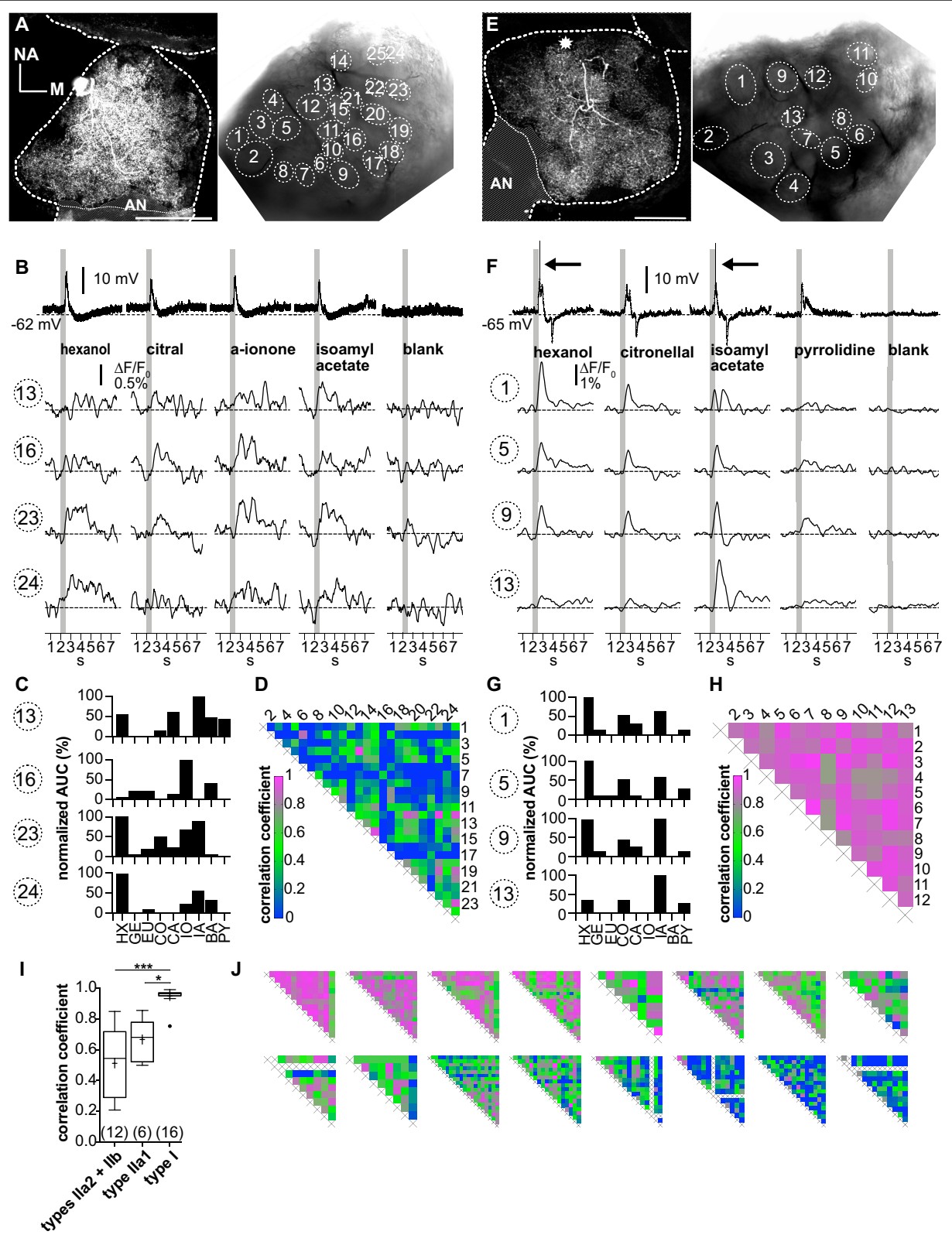

**Figure 5.** Ca²⁺ imaging of type II local interneurons (LNs) shows heterogeneous glomerular odorant responses. Data from a type IIb (**A–D**, LNII #2) and a type IIa LN (**E–H**, LNII #3). (**A, E**) Left: biocytin/streptavidin stainings of the investigated type II LNs. The antennal lobes (ALs) are outlined by the dotted lines. The position of the soma in (**E**) is marked by the star. Scale bar: 100 μm. Right: transmitted light images of the same ALs during the experiment. Outlined glomeruli were marked as regions of interest and individually analyzed. The orientations of the left and right images are similar.

*Figure 5 continued on next page*

*Figure 5 continued*

(**B, F**) Electrophysiological responses to four odorants (top traces) with the corresponding Ca²⁺ dynamics of four glomeruli that are marked in the images shown in (**A**) and (**E**). Gray bars represent the 500 ms odorant stimuli. (**B–D**) Type IIb LN. (**B**) The neuron responded similarly to the different odorants with graded depolarizations that were followed by slow hyperpolarizations. The time course and amplitude of the corresponding Ca²⁺ signals varied in different glomeruli for the different odorants. (**C**) Tuning curves of glomerular Ca²⁺ signals (for details, see *Figure 3C*). The tuning curves of the different glomeruli varied considerably, while the maximum response was induced by different odorants in the different glomeruli. Some glomeruli were narrowly tuned (e.g., glomerulus 16); others were broadly tuned (e.g., glomerulus 23). (**D**) Heatmap showing the correlations between glomerular tuning curves of every imaged glomerulus. Numbers correspond to glomeruli shown in (**A**). Correlations between glomerular tuning curves were mostly low, with coefficients ranging between 0 and 0.97 (median = 0.23). (**F–H**) Type IIa LN. (**F**) The neuron responded similarly to different odorants with graded depolarizations that could include spikelets (e.g., hexanol, isoamylacetate, arrows mark the spikelets), whereas the time course and amplitude of the corresponding Ca²⁺ signals mostly varied between glomeruli for different odorants. (**G**) Tuning curves of the glomerular Ca²⁺ signals shown in (**F**) (for details, see *Figure 3C*). Groups of glomeruli showed similar tuning curves (e.g., glomeruli 1, 5, and 9), while other glomeruli were individually tuned (e.g., glomerulus 13). (**H**) Heatmap showing correlations between glomerular tuning curves of every imaged glomerulus. Numbers correspond to the glomeruli marked in (**E**). Glomerular tuning curves correlated strongly in a subset of glomeruli, while the correlation was low between other glomeruli. Coefficients ranged between 0.72 and 0.98 (median = 0.87). (**I**) The mean correlation coefficient of spiking type I LNs (0.95 ± 0.06, N = 16) is significantly larger compared to nonspiking type IIa1 LNs that can generate spikelets (0.67 ± 0.12, N = 6, p=0.01) and (pooled) type IIa2 and type IIb LNs, which do not generate spikelets (0.51 ± 0.23, N = 12, p<0.001) (Kruskal–Wallis and Dunn's multiple comparisons tests). Abbreviations as in *Figure 3F*. The mean correlation coefficient across all investigated type II LNs was 0.56 ± 0.21 (N = 18). *p<0.05, ***p<0.001. N values are given in brackets. (**J**) Heatmaps of correlations between glomerular tuning curves from all additional type II LNs in descending order of mean correlation coefficient (upper row: LN II # 5, 4, 1, 7, 6, 9, 8, 10; lower row: LN II # 14, 12, 11, 13, 15, 16, 17, 18). AN: antennal nerve, all other abbreviations as in *Figure 3F*.

The online version of this article includes the following figure supplement(s) for figure 5:

**Source data 1.** Numerical data for *Figure 5I*.

**Figure supplement 1.** Glomerular tuning curves of all imaged glomeruli.

individual LNs. In line with the electrophysiological properties, we found odorant-evoked Ca²⁺ signals that were homogeneous across the whole cell in spiking type I LNs and odor and glomerulus-specific Ca²⁺ signals in nonspiking type II LNs. This is also reflected in the highly correlated tuning curves in type I LNs and low correlations between tuning curves in type II LNs. In the following, we discuss whether and how this is consistent with previous studies, suggesting that spiking type I LNs play a role in lateral, interglomerular signaling and why this study assigns a role to nonspiking LNs in local, intraglomerular signaling.

## Interglomerular signaling via spiking type I LNs

Processing of olfactory information in the AL involves complex interactions between the glomerular pathways and between different AL neurons. Previous studies in different insect species have suggested that GABAergic LNs can mediate lateral inhibition by providing inhibitory synaptic input to defined odor-specific arrays of glomeruli. This hypothesis agrees with the current study, where spiking type I LNs showed odorant-specific glomerular Ca²⁺ dynamics, which were always uniform in every imaged glomerulus as long as the generation of Na⁺-driven action potentials was not suppressed. Thus, it is plausible and likely that synaptic inputs to one or a few glomeruli are integrated and result in the firing of action potentials, which propagate to the neurites of all innervated glomeruli where they induce highly correlated voltage-activated Ca²⁺ signals. Since type I LNs express GABA-like immunoreactivity and provide inhibitory input to uPNs and other LNs in all innervated glomeruli (*Distler, 1989*; *Distler and Boeckh, 1997b*; *Husch et al., 2009a*; *Warren and Kloppenburg, 2014*), these neurons are likely part of an inhibitory network that mediates lateral inhibition and contrast enhancement (*Sachse and Galizia, 2002*; *Wilson and Laurent, 2005*). While the current study assigns a role to type I LNs in interglomerular signaling, it does not rule out the possibility that these neurons are also involved in intraglomerular signaling.

In this context, the experiments with suppressed action potential firing reveal an additional functional aspect of type I LNs. They indicate that the hypothesized polar organization of type I LNs with strictly defined (and separated) input and output glomeruli is not entirely correct. When type I LNs were prevented from spiking by hyperpolarization or intracellular block of Na⁺ channels, we observed distinct, glomeruli-specific Ca²⁺ dynamics during odor stimulation. It is likely that these signals originate from Ca²⁺ influx through Ca²⁺ permeable excitatory receptors such as cholinergic receptors, suggesting that each neuron can potentially receive excitatory olfactory input in many innervated glomerulus. This notion agrees with previous studies in the fly, which suggested that individual

GABAergic LNs receive broad, but not uniform, spatial patterns of excitation by either OSNs or PNs (*Wilson and Laurent, 2005*).

## Inter- and intraglomerular signaling via nonspiking type II LN

The nonspiking LNs that have been described in the AL of insects typically innervate all glomeruli (*Husch et al., 2009a*; *Husch et al., 2009b*; *Tabuchi et al., 2015*). In most of these neurons, we observed highly heterogeneous $Ca^{2+}$ dynamics between the individual glomeruli resulting in distinct tuning curves for the individual glomeruli. These heterogeneous and glomerulus-specific $Ca^{2+}$ dynamics imply that type II LNs have distinct functional domains that are (more or less) independent from each other. Accordingly, most nonspiking type II LNs might contribute to microcircuits within glomeruli and mediate intraglomerular signaling rather than interconnecting multiple glomeruli.

Intraglomerular circuits are known from the mouse or rat olfactory bulb (for review, see *Ennis et al., 2015*), where periglomerular, external tufted, and short axon cells interact to modulate the output of Mitral/Tufted cells (*Aungst et al., 2003*; *Liu et al., 2016*; *Najac et al., 2015*; *Wachowiak and Shipley, 2006*). Periglomerular cells are uniglomerular LNs that mediate intraglomerular synaptic signaling. Unlike periglomerular cells, cockroach type II LNs innervate all glomeruli. Still, as these LNs have only weak active membrane properties, postsynaptic potentials just spread within the same glomerulus. Therefore, these neurons could serve similar purposes, and few omniglomerular type II LNs could perform similar functions as many PG uniglomerular cells.

In addition, it is important to consider that nonspiking type II LNs are not a homogenous neuron population (*Fusca et al., 2013*; *Husch et al., 2009b*). In a subpopulation of type II LNs, we observed correlated $Ca^{2+}$ dynamics in subsets of glomeruli. In principle, this could be caused by various mechanisms, including simultaneous input to multiple glomeruli or interglomerular signaling. Note that subpopulations of type II LNs responded to odorant stimulations with strong depolarizations, including spikelets, which apparently can propagate, at least to some extent, to a set of glomeruli. Since this subpopulation of nonspiking type II LNs (type IIa1 LNs) was previously shown to be cholinergic (*Fusca et al., 2013*; *Neupert et al., 2018*), they are likely excitatory. The intrinsic electrophysiological properties of the cholinergic type IIa LNs suggest that they might be part of an excitatory network, which activates neurons in specific sets of glomeruli. This hypothesis is in line with previous studies in the fruit fly, where excitatory LNs, while being multiglomerular, only activate specific glomeruli, thereby providing distinct arrays of glomeruli with excitatory input and distributing odor-evoked activity over an ensemble of PNs (*Das et al., 2017*; *Huang et al., 2010*; *Olsen et al., 2007*; *Root et al., 2007*; *Shang et al., 2007*; reviewed in *Wilson, 2013*).

While type IIa1 are cholinergic and type II LNs generally express multiple neuropeptides (*Fusca et al., 2015*; *Neupert et al., 2012*; *Neupert et al., 2018*), the primary transmitter of most type II LNs is yet to be revealed. One candidate is glutamate, an inhibitory transmitter in the *Drosophila* AL (*Liu and Wilson, 2013*) and cockroach metathoracic motor neurons (*Sattelle, 1992*).

We conclude that in the cockroach AL sensory inputs are processed and computed in inter- and intraglomerular circuits, which are formed by spiking type I and nonspiking type II LNs.

# Materials and methods

**Key resources table**

| Reagent type (species) or resource | Designation | Source or reference | Identifiers | Additional information |
|---|---|---|---|---|
| Chemical compound, drug | Lidocaine N-ethyl chloride | Alomone | QX-314; #Q-150 | Channel blocker |
| Software, algorithm | Prism | GraphPad | RRID:SCR_002798 | |
| Software, algorithm | ImageJ | https://imagej.net/software/fiji/ | RRID:SCR_003070 | |
| Other | Oregon Green 488 BAPTA-1 hexapotassium salt | Thermo Fisher Scientific | Cat. # O6806 | Calcium indicator |

## Animals and materials

*P. americana* were reared in crowded colonies at 27 °C under a 13:11 hr light/dark photoperiod regimen on a diet of dry rodent food, oatmeal, and water. The experiments were performed with adult

males. Unless stated otherwise, all chemicals were obtained from Applichem (Darmstadt, Germany) or Sigma-Aldrich (Taufkirchen, Germany) and had the purity level 'pro analysis.

## Intact brain preparation

The brain preparation leaving the entire olfactory network intact has been described previously (*Demmer and Kloppenburg, 2009*; *Husch et al., 2009a*; *Kloppenburg et al., 1999*). Animals were anesthetized by $CO_2$, placed in a custom-built holder, and the head was immobilized with tape (tesa ExtraPower Gewebeband, Tesa, Hamburg, Germany). The head capsule was opened by cutting a window between the two compound eyes and the antennae's bases. The brain with its antennal nerves and attached antennae was dissected in extracellular saline (see below) and pinned in a Sylgard-coated (Dow Corning Corp., Midland, MI) recording chamber. To get access to the recording site, we desheathed parts of the AL using fine forceps, and preparations were enzymatically treated with a combination of papain (0.3 mg·ml$^{-1}$, P4762, Sigma) and L-cysteine (1 mg·ml$^{-1}$, 30090, Fluka) dissolved in extracellular saline (~3 min, room temperature [RT], ~24 °C). For electrophysiological recordings, the somata of the AL neurons were visualized with a fixed stage upright microscope (AxioExaminer, Carl Zeiss, Jena, Germany) using a 20× water-immersion objective (20× W Apochromat, NA = 1) with a 4× magnification changer, and infrared differential interference contrast optics (*Dodt and Zieglgänsberger, 1994*).

## Identification of AL neurons

The prerequisite to study the physiology of identified neurons is the unequivocal identification of neuron types. The identification was performed as described by *Fusca et al., 2013*. Briefly, AL neurons were first preidentified by the size and location of their somata. Recordings were performed under visual control from cell bodies in the ventrolateral somata group (VSG, *Distler, 1989*), where different neuron types are located in separated clusters. This preidentification has a high success rate for the major neuron types (>90%) and was verified in each case by the physiological and morphological characterization during and after the recording using the following criteria: two main LN types were identified by their distinctive physiological properties: (1) spiking type I LNs that generated Na$^+$-driven action potentials upon odor stimulation and (2) nonspiking type II LNs, in which odor stimulation evoked depolarizations, but no Na$^+$-driven action potentials (*Husch et al., 2009a*; *Husch et al., 2009b*). Type I LNs had arborizations in many, but not all, glomeruli. The density of processes varied between glomeruli of a given type I LNs. Type II LN had processes in all glomeruli. The density and distribution of arborizations were similar in all glomeruli of a given type II LN, but varied between different type II LN. Two subtypes (type IIa and type IIb) can be distinguished by the branch patterns within the glomeruli, the size and branch pattern of low-order neurites, odor responses, and active membrane properties (*Husch et al., 2009b*). Type IIa LNs had strong Ca$^{2+}$-dependent active membrane properties and responded with odor-specific elaborate patterns of excitation and periods of inhibition. In a subset of the type IIa neurons, which are cholinergic (type IIa1, *Fusca et al., 2013*), the excitation included Ca$^{2+}$-driven 'spikelets' riding on the depolarization. In contrast, type IIb LNs responded mostly with sustained, relatively smooth depolarizations.

## Whole-cell recordings

Whole-cell recordings were performed at RT following the methods described by *Hamill et al., 1981*. Electrodes with tip resistances between 2 and 3 MΩ were fashioned from borosilicate glass (inner diameter 0.86 mm, outer diameter 1.5 mm, GB150-8P, Science Products, Hofheim, Germany) with a vertical pipette puller (PP-830 or PC-10, Narishige, Japan). Recording pipettes were filled with intracellular solution containing (in mM): 218 K-aspartate, 10 NaCl, 2 MgCl$_2$, 10 HEPES, and 0.8 Oregon Green 488 BAPTA-1 hexapotassium salt (OGB1, O6806, Thermo Fisher Scientific, Waltham, MA) adjusted to pH 7.2 with KOH. In some experiments, 2 mM lidocaine N-ethyl chloride (QX-314, #Q-150, Alomone, Jerusalem, Israel) was added to the intracellular solution. During the experiments, if not stated otherwise, the cells were superfused constantly with extracellular solution containing (in mM): 185 NaCl, 4 KCl, 6 CaCl$_2$, 2 MgCl$_2$, 10 HEPES, 35 D-glucose. The solution was adjusted to pH 7.2 with NaOH.

Whole-cell current-clamp recordings were made with an EPC10 patch-clamp amplifier (HEKA-Elektronik, Lambrecht, Germany) controlled by the program Patchmaster (version 2.53, HEKA-Elektronik) running under Windows. The electrophysiological data were sampled at 10 kHz. The

recordings were low-pass filtered at 2 kHz with a 4-pole Bessel-Filter. Compensation of the offset potential and capacitive currents was performed using the 'automatic mode' of the EPC10 amplifier. Whole-cell capacitance was determined by using the capacitance compensation (C-slow) of the amplifier. The liquid junction potential between intracellular and extracellular solution was also compensated (16.9 mV, calculated with Patcher's-Power-Tools plug-in [https://www3.mpibpc.mpg.de/groups/neher/index.php?page=aboutppt] for Igor Pro 6 [Wavemetrics, Portland, Oregon]). Voltage errors due to series resistance ($R_S$) were minimized using the RS-compensation of the EPC10. $R_S$ was compensated between 60% and 70% with a time constant ($\tau$) of 10 µs.

## Odor stimulation

To deliver the odorants, we used a continuous airflow system. Carbon-filtered, humidified air was guided across the antenna at a flow rate of ~2 l·min$^{-1}$ ('main airstream') through a glass tube (inner diameter 10 mm) that was placed perpendicular to and within 20–30 mm distance of the antennae. To apply odorants, 5 ml of odorant-containing solutions (either pure or diluted in mineral oil; M8410, Sigma) were transferred into 100 ml glass vessels. Strips of filter paper in the odorant solution were used to facilitate evaporation. The concentration of each odorant was adjusted to match the vapor pressure of the odorant with the lowest value (eugenol). Dilutions were as follows: α-ionone 40.9% (I12409, Aldrich), ±citral 24.2% (C83007, Aldrich), 1-hexanol 2.4% (52830, Fluka), benzaldehyde (418099, Aldrich) 2.2%, citronellal 8.7% (C2513, Aldrich), eugenol 100% (E51791, Aldrich), geraniol 73.7% (48799, Fluka), isoamylacetate (112674, Aldrich) 13.7%, and pyrrolidine 0.035% (83241, Fluka). The headspace of pure mineral oil was the control stimulus (blank). During a 500 ms stimulus, ~17 ml of the vessel volume was injected into the main air stream. The solenoids were controlled by the D/A-interface of the EPC10 patch-clamp amplifier and the Patchmaster software. Odorant-containing air was quickly removed from the experimental setup with a vacuum funnel (inner diameter 3.5 cm) placed 5 cm behind the antennae. To allow for sensory recovery, consecutive odorant stimulations of the same preparation were performed after intervals of at least 60 s with non-odorant containing air.

## Calcium imaging

Odor-evoked calcium dynamics were measured with the Ca$^{2+}$ indicator OGB1 (see the intracellular solution), a single wavelength, high-affinity dye suitable to monitor fast intracellular Ca$^{2+}$ signals. The imaging setup consisted of a Zeiss AxioCam/MRm CCD camera with a 1388 × 1040 chip and a Polychromator V (Till Photonics, Gräfelfing, Germany) that was coupled via an optical fiber into the Zeiss AxioExaminer upright microscope. The camera and polychromator were controlled by the software Zen pro, including the module 'Physiology' (2012 blue edition, Zeiss). After establishing the whole-cell configuration, neurons were held in current-clamp mode, and a hyperpolarizing current (~−200 pA) was injected for about 45–60 min to allow for dye loading. After loading, up to nine different odorants were applied as 500 ms pulses onto the ipsilateral antenna. Odor-induced Ca$^{2+}$ transients in the OGB1-loaded neurons were monitored by images acquired at 488 nm excitation with 50 ms exposure time and a frame rate of ~18 Hz. The emitted fluorescence was detected through a 500–550 nm bandpass filter (BP525/50), and data were acquired using 5 × 5 on-chip binning. Images were recorded in arbitrary units (AU) and analyzed as 16-bit grayscale images.

## Analysis of odor-evoked calcium signals

The analysis was performed offline using ImageJ (version 2.0.0-rc-64/1.51 s) and Prism 7 or 9 (GraphPad, CA). Amplitudes and kinetics of the Ca$^{2+}$ signals were calculated as means (in AU) of individual innervated glomeruli, which were defined as the respective regions of interest (ROIs). ROIs were defined on transmitted light images of the investigated ALs. The Ca$^{2+}$ signals are given as relative fluorescence changes ($\Delta F/F_0$). To correct for bleaching, biexponential fits to the time courses of the glomerular Ca$^{2+}$ signals during the blank stimulus, which lacked the odorant-evoked Ca$^{2+}$ influx, were used.

For statistical analysis of data obtained for the different cell types, nonparametric Kruskal–Wallis tests with Dunn's multiple comparisons tests were performed in Prism 7 or 9. Correlation coefficients from matrices of glomerular tuning curves were calculated with a confidence interval of 95% and are given as nonparametric Spearman correlation r. For the heatmaps representing the correlation coefficients, negative r values were set to zero. A significance level of 0.05 was accepted for all tests. All calculated values are expressed as mean ± standard deviation. In the box plots, horizontal lines show

the median of the data. The boxes indicate the 25th and 75th percentiles. The lower and upper whiskers were calculated according to the 'Tukey' method. *p<0.05, **p<0.01. ***p<0.001. N values are given in brackets. The numerical data for the box plots are given in the Source data for the respective figures. Individual data of each recorded neurons are given in Individual Neurons-*Source data 1*.

## Single-cell and double-labeling and confocal microscopy

To label individual cells, 1% (w/v) biocytin (B4261, Sigma) was added to the pipette solution. After the electrophysiological recordings, the brains were fixed in Roti-Histofix (P0873, Carl Roth, Karlsruhe, Germany) overnight at 4 °C. Subsequently, the brains were rinsed in 0.1 M phosphate buffered saline (PBS, 3 × 20 min and then for ~12 hr, RT). PBS contained (in mM) 72 $Na_2HPO_4 \times 2H_2O$, 28 $NaH_2PO_4 \times H_2O$, resulting in pH 7.2. To facilitate streptavidin penetration, the samples were treated with a commercially available collagenase/dispase mixture (1 $mg \cdot ml^{-1}$, 269638, Roche Diagnostics, Mannheim, Germany) and hyaluronidase (1 $mg \cdot ml^{-1}$, H3506, Sigma-Aldrich) in PBS (1 hr, 37 °C), rinsed in PBS (3 × 10 min, 4 °C) and then preincubated in blocking solution, consisting of PBS containing 1% (w/v) Triton X-100 (A1388, AppliChem) and 10% (v/v) normal goat serum (S-1000, Vector Labs, Burlingame, CA) for 1 hr at RT. The brains were then incubated with *Alexa 633* conjugated streptavidin (1:400, S21375, Invitrogen, Eugene, OR) in PBS supplemented with 10% (v/v) normal goat serum for ~12 hr at 4 °C, rinsed in PBS (3 × 10 min, RT), dehydrated, cleared, and mounted in methylsalicylate.

In some preparations, we used immunohistochemistry to label synapsin to mark the glomeruli. After preincubation in blocking solution and before the streptavidin incubation, these brains were incubated for 5 days at 4 °C with a monoclonal primary mouse antibody against the presynaptic vesicle protein synapsin I (3C11, supernatant; obtained from the Developmental Studies Hybridoma Bank, University of Iowa, RRID:AB_528479), diluted 1:50 in blocking solution. Subsequently, the brains were rinsed in PBS-1% Triton X-100 (2 × 2 hr, RT), incubated in *Alexa 488* conjugated goat anti-mouse secondary antibody for 5 days at 4 °C (1:200 in blocking solution, 115-545-062, Dianova, Hamburg, Germany) and rinsed in PBS-1% Triton X-100 (2 × 2 hr, RT) and PBS (3 × 10 min, RT). 3C11 (anti-SYNORF1) was deposited to the DSHB by Buchner, E. (DSHB Hybridoma Product 3C11 [anti SYNORF1, *Klagges et al., 1996*]).

Fluorescence images were captured with confocal microscopes equipped with Plan-Apochromat 10× (numerical aperture 0.45) and Plan-Apochromat 20× (numerical aperture 0.75) objectives (LSM 510, Zeiss) or with a 20× objective (SP8, Leica Microsystems, Wetzlar, Germany), respectively. *Alexa 633* was excited at 633 nm, and emission was collected through a 650 nm long-pass filter. *Alexa 488* was excited at 488 nm, and emission was collected through a 505–530 nm bandpass filter. Confocal images were adjusted for contrast and brightness and overlaid in ImageJ. The final figures were prepared in Affinity Designer (Serif, Nottingham, UK).

## Acknowledgements

We thank Helmut Wratil for his excellent technical assistance. The work in PK's laboratory was supported by grants from the Deutsche Forschungsgemeinschaft (EXC 2030-390661388 and KL 762/6-1).

## Additional information

### Funding

| Funder | Grant reference number | Author |
| --- | --- | --- |
| Deutsche Forschungsgemeinschaft | EXC 2030-390661388 | Peter Kloppenburg |
| Deutsche Forschungsgemeinschaft | KL 762/6-1 | Peter Kloppenburg |

The funders had no role in study design, data collection and interpretation, or the decision to submit the work for publication.

## Author contributions
Debora Fusca, Conceptualization, Data curation, Formal analysis, Investigation, Writing - original draft, Writing - review and editing; Peter Kloppenburg, Conceptualization, Funding acquisition, Project administration, Resources, Writing - original draft, Writing - review and editing

## Author ORCIDs
Peter Kloppenburg  http://orcid.org/0000-0002-4554-404X

## Decision letter and Author response
Decision letter https://doi.org/10.7554/eLife.65217.sa1
Author response https://doi.org/10.7554/eLife.65217.sa2

## Additional files

### Supplementary files
- Transparent reporting form
- Source data 1. Individual Neurons.

### Data availability
Source data files are provided for Figures 2-5. An additional source data file associated with the whole article contains individual data for each recorded neuron.

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
