## [Decision Letter]

**Acceptance summary:**

This study addresses the computational significance of different types of inhibitory local interneurons (LNs) in olfactory circuits, an interesting question of significant general interest. Using in vivo physiology and calcium imaging of odorant responses, the work provides direct experimental support for distinct roles for spiking and non-spiking LNs present in the antennal lobe of the cockroach Periplaneta Americana. The authors' overall conclusion is that in type I spiking LNs, odors evoke calcium signals globally and relatively uniformly across glomeruli in the antennal lobe and LN neurites in each glomerulus are broadly tuned to odor. In contrast, in type II nonspiking LNs, the authors report odor-specific patterns of calcium signals, with LN neurites in different glomeruli displaying distinct local odor tuning. These results have implications for how inhibitory systems work in other olfactory systems as well as other brain regions.

**Decision letter after peer review:**

Thank you for submitting your article "Task-specific roles of local interneurons for inter- and intraglomerular signaling in the insect antennal lobe." for consideration by *eLife*. Your article has been reviewed by 3 peer reviewers, including Mani Ramaswami as the Reviewing Editor and Reviewer #1, and the evaluation has been overseen by Ronald Calabrese as the Senior Editor.

Essential revisions:

1. A major issue that needs to be addressed pertains to the different amplitudes of calcium signals observed in non-spiking and spiking LNs. Strong signals are naturally expected to be more visible across planes and across glomeruli compared to weak ones – and so the concentration of odor stimuli needs to adjusted to better match the magnitude of calcium signals between cell types that are being directly compared. In its current form, it remains possible that the conclusion for global- versus odor-specific signals will not hold when average LN calcium activity is matched.

(Obviously two-photon imaging, which can provide better optical sectioning and alleviate biases introduced by out of plane signals would improve the study, but this would require easy access to such equipment)

2. The use of an odorant that is known to narrowly activate just one or two glomeruli based on OSN innervation patterns could potentially help to clarify the roles of intra- versus interglomerular inhibition in each LN type. if an odor elicits LN calcium signals in only a subset of glomeruli, it's hard to know if those signals arise from direct local input to those glomeruli (purely intraglomerular inhibition), or whether ORN input to one glomerulus is eliciting calcium signals in another glomerulus (selective interglomerular inhibition). This would most easily clarified by comparing glomerular patterns of ORN input and LN activation. Alternatively, by using a odor that is mostly selective for a single glomerulus, one could evaluate the degree of signal spread in the LN neurite arbor to determine if LN signals are locally confined to the input glomerulus, as would be expected if the LNs function in intraglomerular inhibition. Perhaps such could be identified from knowledge of the chemical ecology of the cockroach? If this is not done, the authors may hypothesize that a function of nonspiking LNs is intraglomerular inhibition, but cannot 100% rule out selective interglomerular inhibition (which is potentially even more interesting). Thus, the language should be tempered down and duly qualified to acknowledge this possibility. (This may serve the authors well down the line if selective interglomerular inhibition ends up also being involved)

3. The Results sections should also be enhanced to better clarify the hypotheses being tested and the likely interpretations of the observations.

---

## [Author Response]

Essential revisions:1. A major issue that needs to be addressed pertains to the different amplitudes of calcium signals observed in non-spiking and spiking LNs. Strong signals are naturally expected to be more visible across planes and across glomeruli compared to weak ones – and so the concentration of odor stimuli needs to adjusted to better match the magnitude of calcium signals between cell types that are being directly compared. In its current form, it remains possible that the conclusion for global- versus odor-specific signals will not hold when average LN calcium activity is matched.(Obviously two-photon imaging, which can provide better optical sectioning and alleviate biases introduced by out of plane signals would improve the study, but this would require easy access to such equipment)

Thank you for allowing us to clarify this point. The reviewer comment implies that the different amplitudes of the Ca^2+^ signals indicate some technical-methodological deficiency (poorly chosen odor concentration). But in fact, this is a key finding of this study that is physiologically relevant and crucial for understanding the function of the neurons studied. These very differences in the Ca^2+^ signals are evidence of the different roles these neurons play in AL. The different signal amplitudes directly show the distinct physiology and Ca^2+^ sources that dominate the Ca^2+^ signals in type I and type II LNs. Accordingly, it is impractical to equalize the magnitude of Ca^2+^ signals under physiological conditions by adjusting the concentration of odor stimuli.

In the following, we address these issues in more detail:

1) Imaging Method

2) Odorant stimulation

3) Cell type-specific Ca^2+^ signals

1) Imaging Method:

Of course, we agree with the reviewer comment that out-of-focus and out-of-glomerulus fluorescence can potentially affect measurements, especially in widefield optical imaging in thick tissue. This issue was carefully addressed in initial experiments. In type I LNs, which innervate a subset of glomeruli, we detected fluorescence signals, which matched the spike pattern of the electrophysiological recordings 1:1, only in the innervated glomeruli. In the not innervated ROIs (glomeruli), we detected no or comparatively very little fluorescence, even in glomeruli directly adjacent to innervated glomeruli.

To illustrate this, (Author response image 1) shows measurements from an AL in which an *uniglomerular projection neuron* was investigated in a set of experiments that were not directly related to the current study. In this experiment, a train of action potential was induced by depolarizing current. The traces show the action potential induced fluorescent signals from the innervated glomerulus (glomerulus #1) and the directly adjacent glomeruli.

**Author response image 1. sa2fig1:** Simultaneous electrophysiological and optophysiological recordings of a uniglomerular projection using the ratiometric Ca^2+^ indicator fura-2. The projection neuron has its arborization in glomerulus 1. The train of action potentials was induced with a depolarizing current pulse (grey bar).

These results do not entirely exclude that the large Ca^2+^ signals from the innervated LN glomeruli may include out-of-focus and out-of-glomerulus fluorescence, but they do show that the bulk of the signal is generated from the recorded neuron in the respective glomeruli.

2) Odorant Stimulation:

It is important to note that the odorant concentration cannot be varied freely. For these experiments, the odorant concentrations have to be within a 'physiologically meaningful' range, which means: On the one hand, they have to be high enough to induce a clear response in the projection neurons (the antennal lobe output). On the other hand, however, the concentration was not allowed to be so high that the ORNs were stimulated nonspecifically. These criteria were met with the used concentrations since they induced clear and odorant-specific activity in projection neurons.

3) Cell type-specific Ca^2+^ signals:

The differences in Ca^2+^ signals are described and discussed in some detail throughout the text (e.g., page 6, lines 119-136; page 9, lines 193-198; page 10-11, lines 226-235; page 14-15, line 309-333). Briefly: In spiking type I LNs, the observed large Ca^2+^ signals are mediated mainly by voltage-depended Ca^2+^ channels activated by the Na^+^-driven action potential's strong depolarization. These large Ca^2+^ signals mask smaller signals that originate, for example, from excitatory synaptic input (i.e., evoked by ligand-activated Ca^2+^ conductances). Preventing the firing of action potentials can unmask the ligand-activated signals, as shown in Figure 4.

In nonspiking type II LNs, the action potential-generated Ca^2+^ signals are absent; accordingly, the Ca^2+^ signals are much smaller. In our model, the comparatively small Ca^2+^ signals in type II LNs are mediated mainly by (synaptic) ligand-gated Ca^2+^ conductances, possibly with contributions from voltage-gated Ca^2+^ channels activated by the comparatively small depolarization (compared with type I LNs).

Accordingly, our main conclusion, that spiking LNs play a primary role in interglomerular signaling, while nonspiking LNs play an essential role in intraglomeular signaling, can be DIRECTLY inferred from the differences in odorant induced Ca^2+^ signals alone.

a) Type I LN: The large, simultaneous, and uniform Ca^2+^ signals in the innervated glomeruli of an individual type I LN clearly show that they are triggered in each glomerulus by the propagated action potentials, which conclusively shows lateral interglomerular signal propagation.

b) Type II LNs: In the type II LNs, we observed relatively small Ca^2+^ signals in single glomeruli or a small fraction of glomeruli of a given neuron. Importantly, the time course and amplitude of the Ca^2+^ signals varied between different glomeruli and different odors. Considering that type II LNs in principle, can generate large voltage-activated Ca^2+^ currents (larger that type I LNS; page 4, lines 82-86, Husch et al. 2009a,b; Fusca and Kloppenburg 2021), these data suggest that in type II LNs electrical or Ca^2+^ signals spread only within the same glomerulus; and laterally only to glomeruli that are electrotonically close to the odorant stimulated glomerulus.

Taken together, this means that our conclusions regarding inter- and intraglomerular signaling can be derived from the simultaneously recorded amplitudes and the dynamics of the membrane potential and Ca^2+^ signals alone. This also means that although the correlation analyses support this conclusion nicely, the actual conclusion does not ultimately depend on the correlation analysis. We had (tried to) expressed this with the wording, “Quantitatively, this is reflected in the glomerulus-specific odorant responses and the diverse correlation coefficiiants across…” (page 10, lines 216-217) and “ …This is also reflected in the highly correlated tuning curves in type I LNs and low correlations between tuning curves in type II LNs”(page 13, lines 293-295).

2. The use of an odorant that is known to narrowly activate just one or two glomeruli based on OSN innervation patterns could potentially help to clarify the roles of intra- versus interglomerular inhibition in each LN type. if an odor elicits LN calcium signals in only a subset of glomeruli, it's hard to know if those signals arise from direct local input to those glomeruli (purely intraglomerular inhibition), or whether ORN input to one glomerulus is eliciting calcium signals in another glomerulus (selective interglomerular inhibition). This would most easily clarified by comparing glomerular patterns of ORN input and LN activation. Alternatively, by using a odor that is mostly selective for a single glomerulus, one could evaluate the degree of signal spread in the LN neurite arbor to determine if LN signals are locally confined to the input glomerulus, as would be expected if the LNs function in intraglomerular inhibition. Perhaps such could be identified from knowledge of the chemical ecology of the cockroach? If this is not done, the authors may hypothesize that a function of nonspiking LNs is intraglomerular inhibition, but cannot 100% rule out selective interglomerular inhibition (which is potentially even more interesting). Thus, the language should be tempered down and duly qualified to acknowledge this possibility. (This may serve the authors well down the line if selective interglomerular inhibition ends up also being involved)

1) Not all nonspiking type II LNs are inhibitory (Fusca et al., 2013; Neupert et al., 2018; page 4, line 89-91; page 16, lines 365-367). Therefore, 'intra- or interglomerular signaling' seems more appropriate instead of 'intra- and interglomerular inhibition'.

2) While we appreciate this comment, we do not see how the suggested experiment can generate 'more conclusive' results, even if an odorant would be available, which "narrowly activates only one or two glomeruli". Glomerular pathways can interact via excitatory LNs (e.g., Das et al., 2017; Olsen et al., 2007; Shang et al., 2007; Yaksi and Wilson, 2010).

3) In contrast to what the reviewer comment implies, we have not ruled the possibility that type II LNs can mediate interglomerular signaling. Actually, this topic has already been discussed in some detail in the original version (page 12, lines 268-271; page 15-16, lines 338-374). This section, in fact, has a respective heading: "Inter- and intraglomerular signal transduction via nonspiking type II LNs" (page 15, line 338). As suggested in the reviewer comment, we have hypothesized intraglomerular signaling as an important task for type II LNs. In addition, we also have acknowledged and discussed in some detail that type II LNs also might be involved in interglomerular signaling. This is discussed especially with regard to the different subtypes of type II neurons (page 16, lines 358-374).

It is important to point out that the issue of intra- versus interglomerular signaling arises only in those experiments in which correlated excitation indeed occurs in multiple glomeruli. In 11 out of 18 type II LNs, we found 'very uncorrelated' (r=0.43±0.16, N=11) glomerular tuning curves. These experiments argue strongly for 'local excitation' with locally restricted intracellular signal propagation and support our interpretation that this group of neurons mediates intraglomerular signaling.

We agree with the reviewer that correlated excitation in multiple glomeruli can, in principle, be caused by direct (parallel) local input in a subset of glomeruli (pure intraglomerular signaling) or by interglomerular signaling. As mentioned above, both possibilities have been discussed in some detail in the original manuscript. In light of the reviewer comments, we have emphasized this point more and adjusted the wording to make it more clear that these alternative mechanisms may exist (page 16, lines 361-363).

3. The Results sections should also be enhanced to better clarify the hypotheses being tested and the likely interpretations of the observations.

We followed the reviewers' suggestions and tried to enhance the result section to clarify our hypotheses and interpretations of the data (page 7, line 157-159; page 9, lines 193-198; page 10, lines 226-235).

References:

Assisi, C., Stopfer, M., Bazhenov, M., 2011. Using the structure of inhibitory networks to unravel mechanisms of spatiotemporal patterning. Neuron 69, 373–386. https://doi.org/10.1016/j.neuron.2010.12.019

Das, S., Trona, F., Khallaf, M.A., Schuh, E., Knaden, M., Hansson, B.S., Sachse, S., 2017. Electrical synapses mediate synergism between pheromone and food odors in *Drosophila melanogaster*. Proc Natl Acad Sci U S A 114, E9962–E9971. https://doi.org/10.1073/pnas.1712706114

Fujiwara, T., Kazawa, T., Haupt, S.S., Kanzaki, R., 2014. Postsynaptic odorant concentration dependent inhibition controls temporal properties of spike responses of projection neurons in the moth antennal lobe. PLOS ONE 9, e89132. https://doi.org/10.1371/journal.pone.0089132

Fusca, D., Husch, A., Baumann, A., Kloppenburg, P., 2013. Choline acetyltransferase-like immunoreactivity in a physiologically distinct subtype of olfactory nonspiking local interneurons in the cockroach (Periplaneta americana). J Comp Neurol 521, 3556–3569. https://doi.org/10.1002/cne.23371

Fuscà, D., and Kloppenburg, P. (2021). Odor processing in the cockroach antennal lobe-the network components. Cell Tissue Res.

Hong, E.J., Wilson, R.I., 2015. Simultaneous encoding of odors by channels with diverse sensitivity to inhibition. Neuron 85, 573–589. https://doi.org/10.1016/j.neuron.2014.12.040

Husch, A., Paehler, M., Fusca, D., Paeger, L., Kloppenburg, P., 2009a. Calcium current diversity in physiologically different local interneuron types of the antennal lobe. J Neurosci 29, 716–726. https://doi.org/10.1523/JNEUROSCI.3677-08.2009

Husch, A., Paehler, M., Fusca, D., Paeger, L., Kloppenburg, P., 2009b. Distinct electrophysiological properties in subtypes of nonspiking olfactory local interneurons correlate with their cell type-specific Ca^2+^ current profiles. J Neurophysiol 102, 2834–2845. https://doi.org/10.1152/jn.00627.2009

Nagel, K.I., Wilson, R.I., 2016. Mechanisms Underlying Population Response Dynamics in Inhibitory Interneurons of the *Drosophila* Antennal Lobe. J Neurosci 36, 4325–4338. https://doi.org/10.1523/JNEUROSCI.3887-15.2016

Neupert, S., Fusca, D., Kloppenburg, P., Predel, R., 2018. Analysis of single neurons by perforated patch clamp recordings and MALDI-TOF mass spectrometry. ACS Chem Neurosci 9, 2089–2096.

Olsen, S.R., Bhandawat, V., Wilson, R.I., 2007. Excitatory interactions between olfactory processing channels in the *Drosophila* antennal lobe. Neuron 54, 89–103. https://doi.org/10.1016/j.neuron.2007.03.010

Olsen, S.R., Wilson, R.I., 2008. Lateral presynaptic inhibition mediates gain control in an olfactory circuit. Nature 452, 956–960. https://doi.org/10.1038/nature06864

Sachse, S., Galizia, C., 2002. Role of inhibition for temporal and spatial odor representation in olfactory output neurons: a calcium imaging study. J Neurophysiol. 87, 1106–17.

Shang, Y., Claridge-Chang, A., Sjulson, L., Pypaert, M., Miesenbock, G., 2007. Excitatory Local Circuits and Their Implications for Olfactory Processing in the Fly Antennal Lobe. Cell 128, 601–612.

Wilson, R.I., 2013. Early olfactory processing in *Drosophila*: mechanisms and principles. Annu Rev Neurosci 36, 217–241. https://doi.org/10.1146/annurev-neuro-062111-150533

Yaksi, E., Wilson, R.I., 2010. Electrical coupling between olfactory glomeruli. Neuron 67, 1034–1047. https://doi.org/10.1016/j.neuron.2010.08.041